# Multi-Domain Feature Alignment for Face Anti-Spoofing

**DOI:** 10.3390/s23084077

**Published:** 2023-04-18

**Authors:** Shizhe Zhang, Wenhui Nie

**Affiliations:** School of Computer Science and Communication Engineering, Jiangsu University, Zhenjiang 212013, China; zsz2021@stmail.ujs.edu.cn

**Keywords:** multi-domain feature alignment domain generalization (MADG), face anti-spoofing, feature alignment, multiple source domain, domain generalization, transfer learning

## Abstract

Face anti-spoofing is critical for enhancing the robustness of face recognition systems against presentation attacks. Existing methods predominantly rely on binary classification tasks. Recently, methods based on domain generalization have yielded promising results. However, due to distribution discrepancies between various domains, the differences in the feature space related to the domain considerably hinder the generalization of features from unfamiliar domains. In this work, we propose a multi-domain feature alignment framework (MADG) that addresses poor generalization when multiple source domains are distributed in the scattered feature space. Specifically, an adversarial learning process is designed to narrow the differences between domains, achieving the effect of aligning the features of multiple sources, thus resulting in multi-domain alignment. Moreover, to further improve the effectiveness of our proposed framework, we incorporate multi-directional triplet loss to achieve a higher degree of separation in the feature space between fake and real faces. To evaluate the performance of our method, we conducted extensive experiments on several public datasets. The results demonstrate that our proposed approach outperforms current state-of-the-art methods, thereby validating its effectiveness in face anti-spoofing.

## 1. Introduction

With the extensive use of deep learning in computer vision, face recognition (FR) [1,2] technology has become increasingly important in daily life, particularly in scenarios that require user identification and authorization. Despite significant progress in FR, these systems remain susceptible to various types of attacks, such as print attacks, replayed video attacks, and 3D mask attacks. To address these challenges, current state-of-the-art research has proposed various methods for face anti-spoofing (FAS) [3,4,5,6]. These methods can be broadly categorized into two groups: hand-crafted feature-based and deep learning feature-based approaches.

Despite the notable achievements of previous face anti-spoofing (FAS) methods in intra-domain testing, their performance significantly deteriorates in cross-domain testing. This is primarily due to the introduction of bias resulting from the distinct characteristics of domains and the inability to address such bias by considering their internal relationships. Consequently, the generalization effect of the model on the novel domain is insufficient. To mitigate this limitation, recent studies have utilized unsupervised-learning-based domain adaptation (DA) techniques to eliminate the domain bias between source and target domains. Nevertheless, the target domain usually denotes an unseen domain for the source domain, and acquiring an adequate amount of target domain data for training in real-world scenarios is not typically feasible.

In response to the weakness of a model’s generalization in the unseen domain, several studies related to domain generalization (DG) have been proposed. Conventional DG [7] proposed a novel multi-adversarial discriminative method to learn a discriminative multi-domain feature space and improve the generalization performance. This method aimed to find a feature space with multiple source domains aligned, assuming that the feature extracted from the unseen domain can map near that feature space, indicating that the model is well generalized over the target domain. However, Conventional DG fails to account for the difference in generalization between real and fake faces. To address this limitation, SSDG [8] proposed an end-to-end single-side method that separates the feature distributions of real and fake faces by pulling all the real faces and pushing the fake faces of different domains, resulting in a higher classification capacity than Conventional DG. However, this method is insufficient in eliminating the interference of domain-related cues in the domain generalization problem. Motivated by the above works, this study aims to align the feature spaces corresponding to multiple source domains while dividing the distribution of fake and real faces. As shown in Figure 1, our proposed method aligns multiple source domains to obtain a more generalizable feature space, outperforming the baseline method in terms of generalization and classification performance.

Based on the aforementioned considerations and an analysis of the feature space of the source domain, we found that differences in features between multiple source domains significantly interfere with the generalization effectiveness of face anti-spoofing (FAS) in an unseen feature space. To address this issue, we introduce a novel framework for domain generalization in this work, named the Multi-domain Feature Alignment Domain Generalization (MADG) framework. Specifically, the proposed framework utilizes feature generators to produce real and fake face features in each domain, which are subsequently aligned using a multi-domain feature alignment method that employs a loss function based on minimized margin discrepancy disparity between multiple source domains. To optimize this loss function, we use multiple adversarial learning processes. The feature alignment process tackles the challenges associated with generalization, while the employment of multi-directional triplet mining strengthens class boundaries to further enhance the classification performance.

The present work makes four primary contributions:We propose a novel source-domain-alignment domain generalization algorithm that utilizes margin disparity discrepancy and adversarial learning. This approach is designed to improve the generalization performance of the FAS model significantly.In the context of multi-domain problems, we devise two alignment strategies and modularize the alignment process. The experimental findings demonstrate that multi-domain alignment surpasses cross-domain alignment in terms of generalization performance, rendering it more advantageous in multi-domain scenarios.We combine the new algorithm with multi-directional triple mining and analyze the source domain feature space. This results in a novel multi-domain feature alignment framework (MADG), which improves the classification accuracy of the FAS model.Extensive experiments and comparisons have been conducted, demonstrating that our proposed approach achieves state-of-the-art performance on most protocols.

The rest of this article is structured into four main sections. In the section second, we provide a detailed introduction to the related work, including mainstream research on face anti-spoofing, domain generalization, and multi-domain learning. We also discuss some works that have provided inspiration for this study. In the third section, we present the proposed method, which includes a comprehensive introduction to the new alignment method and domain generalization framework developed in this study. The fourth section details the experiments conducted to evaluate the proposed method and presents a thorough analysis of the experimental results. Finally, in the fifth section, we provide a summary of our work and give an outlook for future research directions in this area.

## 2. Related Work

### 2.1. Face Anti-Spoofing Methods

There exist two principal categories of conventional face anti-spoofing (FAS) techniques: appearance-based and temporal-based methods. Appearance-based methods involve the extraction of hand-crafted features for classification, such as local binary patterns (LBPs) [9,10] and scale-invariant feature transform (SIFT) [11]. However, temporal-based FAS methods detect attack faces by extracting temporal cues from a sequence of consecutive face frames. Mouth movement detection [12] and eye blink detection [13,14] are examples of the earliest dynamic texture-detection methods. However, these methods do not generalize well to cross-dataset testing scenarios due to the dissimilarities in feature spaces among diverse domains, which often lead to feature bias during generalization.

Recently, deep neural networks, specifically convolutional neural networks (CNNs), have gained widespread adoption in computer vision tasks and have been extensively applied to FAS. Yang et al. [15] were the pioneers in utilizing binary classification CNNs for FAS. Jourabloo et al. [16] proposed a face de-spoofing technique that performs fake face classification by reverse decomposition into real faces and spoof noise. Liu et al. [17] presented a CNN-RNN network that combines both appearance and temporal cues to detect spoof attacks using remote photoplethysmography (rPPG) signals. Similarly, 3D mask detection attack methods [18,19] also exploit rPPG information. Yang et al. [20] combined temporal and appearance cues to detect fake faces from real ones. Roy et al. [21] investigated frame-level FAS to enhance biometric authentication security against face-spoofing attacks. More recently, deep neural networks have been applied to FAS [4,7,8,22,23,24], achieving superior performance compared to conventional methods [15,25,26,27].

In conclusion, traditional FAS methods include appearance-based and temporal-based methods, which extract hand-crafted features and mine temporal cues, respectively. However, deep neural networks, especially CNNs, have achieved state-of-the-art performance in FAS by combining the appearance and temporal cues, using techniques such as reverse decomposition and frame-level FAS for improved biometric authentication security.

### 2.2. Multi-Domain Learning

The use of multiple datasets has recently sparked research interest in multi-domain processing. In particular, the research community has amassed several large-scale FAS datasets with rich annotations [28,29,30,31]. Our work on multi-source domain processing shares similarities with domain adaptation (DA) methods [32,33,34,35,36,37,38,39] that require a retrained model to perform well on both source and target domain data. Specifically, Zhang et al. [37] introduced the concept of margin disparity discrepancy to characterize the differences between source and target domains, which has inspired our work. Ariza et al. [40] conducted a comparative study of several classification methods, which informed our experimental design. Additionally, Liu et al. [41] proposed the YOLOv3-FDL model for successful small crack detection from GPR images using a four-scale detection layer. Notably, the common approaches of Mancini et al. [34] and Rebuffi et al. [35] employ the ResNet [42] architecture, which offers benefits over architectures such as VGG [43] and AlexNet [44] by increasing abstraction through convolutional layers. Yang et al. [45] recently enriched FAS datasets from a different perspective to achieve multi-domain training, while Guo et al. [46] proposed a novel multi-domain model that overcomes the forgetting problem when learning new domain samples and exhibits high adaptability.

### 2.3. Domain Generalization

Domain adaptation (DA) and domain generalization (DG) are two fundamental methods used in FAS research. While the DG method mines the relationships among multiple domains, the DA method aims to adapt the model to a target domain. In this work, we propose a novel domain generalization method that introduces a new loss function inspired by the work of Motiian et al. [47], which encourages feature extraction in similar classes. To align multiple source domains for generalization, previous works such as Ghifary et al. [48] and Li et al. [49] have proposed autoencoder-based approaches. Our method follows a similar approach of learning a shared feature space across multiple source domains that can generalize to the target domain. Previous works such as Shao et al. [7], Saha et al. [50], Jia et al. [8], and Kim et al. [51] have also attempted to achieve this goal. Among them, the single-side adversarial learning method proposed in SSDG [8] is the work most related to ours. However, this end-to-end approach overlooks the relationships between different domains. To address the overfitting and generalization problems of adversarial generative networks, Li et al. [5] proposed a multi-channel convolutional neural network (MCCNN). Additionally, meta-learning formulations [52,53,54] have been utilized to simulate the domain shift during training to learn a representative feature space. However, recent works such as Wang et al. [6] have not adopted a domain-alignment approach but have instead increased the diversity of labeled data by reassembling different styles and content features in their SSAN method.

## 3. Proposed Approach

### 3.1. Overview

The proposed method aims to enhance the generalization ability of face anti-spoofing (FAS) by improving feature space generalization. Generalization to unseen domains is challenging, as samples from the target domain are usually not available during training. However, similarities exist between the features of the source domains and the target domain, and our approach aims to identify these similarities by aligning multiple source domains. To this end, we propose a multi-domain feature alignment framework called MADG, as shown in Figure 2. The feature generator extracts features that are cross-domain aligned using an adversarial learning process called cross-domain alignment (CDA), and multiple CDA modules are combined to form a multi-domain alignment method. Moreover, we introduce a multi-directional triplet mining process to better separate the distribution differences between real and fake faces in the feature space. Thus, MADG can consider the distribution differences between real and fake faces while aligning the feature space of multiple source domains, leading to improved classification performance in new, unseen domains.

### 3.2. Multi-Domain Feature Alignment

This study considers *N* source domains, each with a set of training samples denoted by $X=\{X_{1},X_{2},...,X_{N}\}$, and their corresponding labels, *Y*. In the context of the face anti-spoofing task, the labels correspond to $Y_{r}$ and $Y_{f}$ for real and fake faces, respectively. However, due to significant distribution gaps between different domains, it is necessary to minimize the disparity in feature space. To address this challenge, we propose a multi-domain feature alignment method that consists of two key components: feature generators for generating features and multiple weighted adversarial learning modules combined for multi-domain feature alignment.

**Pre-alignment Feature Generator and Alignment Strategies.** We designed pre-alignment feature generators to transform sample data into features, as follows:(1)$$T_{m}=G_{m}\left(X_{m}\right),T_{s}=G_{s}\left(X_{s}\right),$$

The features extracted from mixed samples of real and fake faces are denoted as $T_{m}$, and their corresponding feature generators as $G_{m}$. Similarly, $T_{s}$ and $G_{s}$ represent the features and generators generated from real or fake faces. These features are known to contain a significant number of domain-relevant clues, which may adversely affect the generalization performance. Before feature alignment, two alignment strategies are devised for the features extracted from real and fake faces as well as mixed samples. In the first strategy, real and fake faces from each domain are independently fed into the respective feature generator. In contrast, the second strategy involves feeding a mixture of real and fake faces from each domain into the feature generator. The two frameworks resulting from these strategies are referred to as MADG-S (single-sided alignment MADG) and MADG-M (mixed alignment MADG), respectively. We compare the outcomes of these two strategies in Section 4.3.4 to determine the optimal method.

**Cross-domain Alignment Module.** To enhance the generalization ability of the feature space for the face anti-spoofing task, it is crucial to reduce the presence of domain-relevant cues in the extracted features. Aligning the features of multiple domains into a common feature space effectively reduces deviations from any specific domain, leading to better performance on unseen domains. This insight is akin to feature alignment techniques employed in domain adaptation.

With the aim of reducing the presence of domain-relevant cues and improving generalization performance, we propose a cross-domain feature alignment module to align the features of different domains. There is typically a substantial gap between the features of different domains, characterized by domain-relevant clues, which can be quantified using measures such as the margin disparity discrepancy (MDD) proposed by Zhang et al. [37]. By progressively reducing this gap, we can obtain an aligned feature space in which domain-specific characteristics become less pronounced. Specifically, our method considers two source domains, *S*1 and *S*2, and leverages the cross-domain alignment module to achieve this goal. MDD has the following key property:(2)$$D_{f,F}^{\left(\rho \right)}\left(S1,S2\right)\geq err_{S2}\left(h\right)-err_{S1}^{\left(\rho \right)}\left(h\right)-\lambda ,$$
where h∈H denotes a label classifier and err represents the error rate of the classifier, while $err^{\left(\rho \right)}$ is the margin error and ρ is the corresponding margin. The scoring function and hypothesis set are denoted by *f* and F, respectively. *D* denotes the margin disparity discrepancy and λ=λ(ρ,F,P,Q) is the ideal combined margin loss. The method proposed by Zhang et al. [37] is a domain adaptation method. err in the above equation can be regarded as the generalization error in domain adaptation. From this inequality, the generalization error of S1 on S2 is upper-bounded, as we can see. According to this property, we only need to optimize this upper bound to reduce this error and achieve the purpose of feature alignment. To this end, we designed an adversarial learning module to align domain features, as depicted in Figure 3. The objective is to minimize MDD, and this process can be formulated as a minimax game, expressed as follows: (3)$$\min_{f,\psi }\varepsilon \left(\hat{S1}\right)+\eta D_{\gamma }\left(\hat{S1},\hat{S2}\right),\max_{f^{\prime }}D_{\gamma }\left(\hat{S1},\hat{S2}\right),$$
where $\hat{S1},\hat{S2}$ are the samples drawn independently from distribution S1,S2, respectively. To strengthen the minimization process, we introduce a feature extractor ψ, and ε denotes cross-entropy loss. *f* and $f^{\prime }$ are classifiers that share the same hypothesis space, and γ denotes the margin factor that is related to the margin of $f^{\prime }$. Through this adversarial learning process, we can obtain the minimized cda between two domains:(4)$$\min_{G}cda_{p,q}=\min_{f,\psi }\max_{f^{\prime }}M\left(S1,S2\right),$$
where *M* denotes the adversarial learning function.

**Multi-domain Alignment Loss.** A multi-domain alignment loss was designed by combining all the obtained cross-domain diverging divergence values *m*, denoted as:(5)$$L_{Mdalign}=\frac{2}{N(N-1)}\sum_{i+j=N,i<j}\lambda_{i,j}m_{i,j},$$
where $\lambda_{i,j}$ is used to control the weight of the alignment of different domain pairs. Given that the disparity between various domains is diverse, choosing the appropriate value for λ can significantly affect the overall alignment effect, and therefore, the generalization effect of the network. We adjusted the value of λ on a specific task protocol (O&M&I to C) for comparison, see Section 4.3.2 for details.

### 3.3. Multi-Directional Triplet Mining

In the context of different domains’ feature spaces, the disparity between real and fake faces is often less pronounced within domains than across them. This is due to the relatively strong similarities between attack samples corresponding to each real sample, which can arise from variations in data collection techniques and attack methods. Our framework aims to learn a concise feature space for the same class. Nonetheless, scattering real faces in a feature space is often easier. To overcome this challenge, we partition the feature spaces of real and fake faces and seek to compress all real faces into a compact feature space. To this end, we utilize a multi-directional triplet loss proposed by SSDG [8]. This loss enables us to mine triplet relationships between features, which can improve the generalized feature space’s performance. The optimization of the loss is expressed as follows: (6)$$\min_{G}L_{Trip}\left(G\right)=\sum_{x_{i}^{i}s,x_{i}^{p},x_{i}^{n}}(∥f\left(x_{i}^{a}\right)-f\left(x_{i}^{p}\right){∥}_{2}^{2}-∥f\left(x_{i}^{a}\right)-f\left(x_{i}^{n}\right){∥}_{2}^{2}+\alpha ),$$

α is the margin we need to define beforehand and $x_{i}^{a}$ is the anchor, while $x_{i}^{p}$ is the positive example and $x_{i}^{n}$ is the negative one.

### 3.4. Loss Function

As shown in Figure 2, our framework employs a face anti-spoofing classifier that combines triplet feature generators after aligning all source domains. The final loss to optimize is a cross-entropy loss based on a hybrid domain with multi-domain alignment and category separation by triplet mining, denoted as $L_{cls}$. Once all components are integrated, the proposed multi-domain feature alignment framework can be obtained as follows:(7)$$L_{MADG}=L_{cls}+\lambda_{1}L_{Mdalign}+\lambda_{2}L_{Trip}$$

The weighting factors $\lambda_{1}$ and $\lambda_{2}$ are leveraged to adjust the balance between the process of aligning multiple domains and separating subclasses between different domains. The values of these factors govern the relative importance of each objective in the overall optimization process.

## 4. Experiments

### 4.1. Experimental Setting

#### 4.1.1. Databases and Protocols

To assess the effectiveness of the proposed approach, we conducted experiments on four publicly available face anti-spoofing datasets; namely, Idiap Replay-Attack [28] (referred to as I), CASIA-FASD [29] (referred to as C), MSU-MFSD [30] (referred to as M), and OULU-NPU [31] (referred to as O). Real and attack face samples from these datasets are shown in Figure 4. Notably, the datasets have a significantly different composition, with variations in display devices, attack types, illumination, and background complexity. For instance, the display devices in I are the iPhone 3Gs and iPad, while C uses the iPad, M utilizes the iPad Air and iPhone 5S, and O deploys the Dell 1905FP and Macbook Retina. Moreover, the attack types vary across the datasets, with I and O containing replayed photo samples, C having cut photo samples, and all datasets including printed photos and replayed videos. The datasets also exhibit differences in illumination, with I and O having extra light, while C and M do not. Additionally, the datasets have varying levels of background complexity, with I, C, and M having a complex background, while O has a less complex background. These differences contribute to significant inter-domain differences in the feature space of the datasets, making them suitable for evaluating the efficacy of the proposed approach.

Therefore, to provide a comprehensive evaluation, we employed the leave-one-out strategy, where one of the databases was designated as the test set, and the remaining three were used as the training set. Thus, four task protocols were established: O&C&I to M, O&M&I to C, O&C&M to I, and I&C&M to O. However, to further examine the method’s effectiveness under various multi-domain scenarios in multi-domain research, additional protocols need to be designed for controlled experiments on the method’s performance. Thus, we extended our experiments to include three domains, similar to the aforementioned four, resulting in a total of 12 task protocols. This comprehensive approach enabled us to analyze and compare the proposed method’s performance in various multi-domain scenarios.

#### 4.1.2. Evaluation Metrics

In order to assess the effectiveness of our model in achieving the desired outcomes, it is crucial to evaluate its performance. To this end, we follow a similar evaluation approach to that of SSDG [8], the most relevant work in this field. The two evaluation metrics we use to assess the performance of our model are HTER and AUC. HTER is calculated as half of the sum of FAR and FRR, which represent the rates of falsely accepting and falsely rejecting a genuine user, respectively. This metric is widely used in liveness detection to measure the model’s ability to distinguish between live and fake faces. However, AUC is a commonly used evaluation metric in classification tasks that measures the area under the ROC curve. The ROC curve plots the true positive rate (TPR) against the false positive rate (FPR) for different classification thresholds. A higher AUC value indicates better classification performance, as the model can achieve a high TPR while maintaining a low FPR. By utilizing these metrics, we can obtain a comprehensive evaluation of the performance of our model and compare it with other state-of-the-art methods.

### 4.2. Implementation Details

#### 4.2.1. Data Preprocessing

To prepare the datasets for our experiments, it was necessary to preprocess both real and fake face images from the original video data. The preprocessing involved extracting frames at random intervals from each video clip and applying the MTCNN [55] method to identify and crop the faces in the resulting images. The cropped faces were then rotated and normalized to create input data of size 256×256×3. This process ensured that the RGB channel was the sole source of information used during training, which simplified the framework’s complexity. The preprocessing step was critical for enabling our model to accurately learn from both the real and fake faces and was essential for ensuring the robustness and reliability of our experimental results.

#### 4.2.2. Network Structure

Our framework was implemented on the platform PyTorch. The details of our network structure are shown in Table 1, Table 2 and Table 3.

**Details of Feature Generator.** To construct the feature generator, we employed ResNet-18 [42] as the backbone. The feature generator was designed by following the ResNet architecture, which includes a convolutional layer with a 7×7 convolution kernel and a max pooling layer at the head. The remaining convolutional layers utilize 3×3 kernels and are accompanied by a batch normalization (BN) layer to improve the stability and speed of network convergence. Some of the convolutional layers use a rectified linear unit (ReLU) activation function, as indicated in the table by conv2, while conv1 lacks the ReLU layer. We chose the ResNet-18 model due to its ability to strike a balance between performance and computational complexity. By incorporating ResNet-18, our feature generator can effectively learn and extract high-level features from facial images.

**Details of Feature Embedder and Classifier.** The residual block utilized in this section features the same structure as the one employed in the feature generator. The classifier comprises a linear model and fully connected layers, with a bottleneck fully connected layer added prior to the classifier.

**Details of Alignment Classifier.** An adversarial learning process is employed in this work to align the source domains, with a network functioning as an alignment classifier. The classifier is composed of multiple components, including a bottleneck layer, a head layer, an adversarial head layer, and a gradient reversal layer (GRL). The bottleneck and head layers consist of fully connected layers, as do the adversarial head layers. During backpropagation, the GRL layer inverts the gradients, enabling the network to learn domain-invariant features.

#### 4.2.3. Training Setting

The optimization of the framework was performed using stochastic gradient descent (SGD) with a momentum of 0.9 and weight decay of 5 × 10${}^{-4}$. The initial learning rate was set to 1 × 10${}^{-2}$ and gradually decreased to 1 × 10${}^{-5}$. To accommodate the limited GPU memory, a batch size of 10 was used for each domain, resulting in a batch size of 30 for all three domains during training. We set the values of $\lambda_{1}$ and $\lambda_{2}$ to 30 and 1, respectively. Furthermore, the $\lambda_{i,j}$ values used to constrain the multi-domain alignment process were all set to 1. These weights are defined in Section 3.

#### 4.2.4. Testing Setting

During the testing phase, the performance of the trained model was assessed on new test samples *x* by feeding them into the model for classification. The classification outcome is denoted as *l* and obtained via l=G(F(x)), where *G* represents the trained model, and *F* refers to the feature generator utilized during training. The classification results on the testing set were used to evaluate the model’s generalization ability, which is a crucial factor for machine learning models. Furthermore, the testing results can be utilized to compare the performance of our proposed method with state-of-the-art methods currently available.

### 4.3. Discussion

#### 4.3.1. Ablation Study

To assess the individual contributions of each component in our proposed framework, we performed various ablation experiments. We report the findings of these experiments in Table 4. Our proposed method is denoted as **MADG**, which includes multi-domain feature alignment loss ($L_{Mdalign}$), multi-directional triplet loss (denoted as $L_{Trip}$), and cross-entropy loss.

The outcomes presented in Table 4 highlight the substantial influence of each constituent of our proposed framework. Ablating any part of the framework led to a conspicuous deterioration in classification performance, which affirms the effectiveness of each component. Specifically, the experiment pertaining to the results without $L_{Trip}$ corresponded to the deactivation of the triplet loss during both forward and backward propagation. The experiment associated with the outcomes without $L_{Mdalign}$ deactivated the multi-domain alignment component.

#### 4.3.2. Comparisons of Different Alignment Weights

The cross-domain feature alignment module is a fundamental component of our proposed multi-domain feature alignment method. By integrating this module into the training process, we can effectively align the features of two source domains. Our multi-domain alignment strategy involves aligning multiple source domains in pairs and controlling these domain pairs by weight. Furthermore, we align multiple source domains simultaneously. For instance, in the scenario involving three source domains, denoted as O, M, and I, we simultaneously align O&M, O&I, and M&I, with each alignment procedure constrained by a weight. To assess the impact of varying weights on the model’s generalization performance, we conducted a series of experiments using the O&M&I to C task protocol. The weights were adjusted using a control variable method, where only one weight λ was varied while the others were held constant at 1. The experimental results, depicted in Figure 5, reveal the impact of the weights on the model’s performance.

By adjusting the weights assigned to the alignment process of each pair of domains, we can evaluate the impact of each domain on both the overall generalization performance and the inter-domain feature characteristics. As depicted in Figure 5, the alignment process of the O&I domain exhibits the most significant impact on the generalization performance. Specifically, as the weight of the O&I alignment process ($\lambda_{OI}$) increases, the generalization performance notably decreases, indicating a significant dissimilarity between the domain-relevant features of O&I and M. However, the optimal generalization performance can be achieved by enhancing the weight of the O&M alignment process ($\lambda_{OM}$).

#### 4.3.3. Experiments on Limited Source Domains

**Limited Source Domain Protocols.** In addition to the leave-one-out experiment, we performed experiments in which we trained the model using a subset of the available source domains. More specifically, we selected two of the three available source domains and used them as the training set. This approach yielded 12 distinct task protocols: O&M to C, O&I to C, M&I to C, C&M to O, C&I to O, M&I to O, C&O to M, C&I to M, I&O to M, C&M to I, C&O to I, and O&M to I. The experimental results of SSDG (baseline), MADG-M (ours), and MADG-S (ours) on these 12 task protocols are shown in Table 5, where the best result for each protocol is shown in bold.

**Analysis of Limited Source Domain Experiments.** The impact of training set diversity and data quantity on the generalization performance of the proposed framework is demonstrated in Table 4 and Table 5. As indicated in Table 5, the generalization performance of the two-source domains is generally lower than that of the three-source domains. Additionally, significant variations in the experimental results of the 12 task protocols are observed (Table 5), which is likely due to differences in data characteristics across domains. Therefore, the domain-relevant clues between the source and target domains may have a lesser effect on some task protocols, yet still attain good generalization performance. Furthermore, a comparison with other state-of-the-art methods on limited source domains is presented in Section 4.4.1. Moreover, Table 5 shows that the limited source domain task protocol is impacted differently by various alignment strategies. The subsequent Section 4.3.4 provides further comparisons of these alignment strategies.

#### 4.3.4. Comparison of Different Alignment Strategies

To provide a comprehensive comparison of the generalization effects of the two alignment strategies implemented in the MADG-M and MADG-S frameworks, we conducted experiments on 12 limited source domain task protocols and 4 leave-one-out task protocols. To ensure a fair evaluation, we compared all of these protocols with the baseline method. The results of the limited source domain experiments are presented in Table 5, while the results of the complete source domain experiments are shown in Table 6.

Based on the experimental results presented in Table 5 and Table 6, it is apparent that our proposed alignment strategies consistently outperform the baseline method in most task protocols. However, the extent of improvement over the baseline is more limited in the limited source domain experiments than in the experiments conducted with three source domains. This finding suggests that our method performs better with a larger number of source domains, indicating greater potential in scenarios with more source domains. Additionally, the alignment effect of our method can be influenced by the distribution of features from the source domains. Therefore, the performance of different alignment strategies for various task protocols can differ, each with its distinct set of strengths and weaknesses, as revealed by the experimental results. These observations are discussed in more detail in Section 4.4.2.

### 4.4. Comparison with State-of-the-Art Methods

#### 4.4.1. Comparison on Limited Source Domains

Our proposed method, as shown in Table 7, outperforms the previous state-of-the-art method SSAN [6] in the restricted source domain task protocols. The objective of SSAN was to enhance generalization by reorganizing content and style features. In comparison, our method achieves a significant improvement by offering a source domain alignment method that improves the generalization performance on the training set of double source domains. Since the multi-domain alignment strategy is not involved in the double source domain alignment, it is easier to train a well-aligned generalized feature space. The method demonstrates considerable advantages in cases of insufficient data, and can maintain good performance even with a small training set, thereby making it valuable for practical applications.

#### 4.4.2. Comparison with Baseline Methods

To assess the effectiveness of our proposed method in improving classification performance, we compared it with the SSDG method [8], a one-sided domain generalization approach that discriminates between real and fake faces to achieve a certain generalization performance. To bring samples from different domains closer together, we incorporated a feature alignment algorithm before triplet mining and combined the domain alignment feature space to improve the aggregation of the aligned real and fake faces. Our method has two different alignment strategies to account for the large differences in source data distribution across different task protocols. The comparison results of SSDG and our method are presented in Table 8.

The SSDG method is considered a strong baseline approach and outperforms most state-of-the-art methods, as demonstrated in Table 8. While SSDG aims to achieve generalization across multiple domains by focusing on domain differences and utilizing asymmetric optimization goals for real and fake faces, it may not effectively capture domain-relevant features. In contrast, our proposed MADG method leverages asymmetric optimization while simultaneously considering alignment across multiple domains. The superior performance of our method, as shown in Table 6, suggests that source domain alignment can lead to improved generalization results. Furthermore, our method’s ability to incorporate multiple alignments accounts for the diverse distribution of source data across different task protocols.

#### 4.4.3. Comparison with Other SOTA Methods

Table 8 presents a comparative analysis of our proposed MADG method with state-of-the-art approaches under the leave-one-out protocols. The best performance in each column is highlighted in bold. The table demonstrates the effectiveness of our approach, as it outperforms all other methods, except for the SSDG [8] approach, which relies on discriminating real faces from fake ones to achieve generalization. In contrast, other methods, such as [7,9,10,15,17,30,31,49], demonstrate significantly poorer performance than our method, perhaps due to their inability to generalize effectively across multiple domains. Although both MADDG [7] and SSDG methods adopt the DG method to learn discriminative cues, the MADDG method struggles to optimize real and fake faces simultaneously, while the SSDG method disregards domain-relevant features. Specifically, SSDG relies on the broader feature distribution extracted from fake faces than real faces, while our MADG approach focuses on feature alignment across multiple domains while considering the distribution discrepancies between fake and real faces. This feature alignment capability enables our approach to achieve superior generalization performance while minimizing the impact of domain-specific features on generalization. Furthermore, different alignment methods may produce varying outcomes for various task protocols, depending on the source domain distribution.

### 4.5. Visualization and Analysis

#### 4.5.1. Feature Visualization

We employed t-SNE visualization, a powerful technique for visualizing high-dimensional data in a low-dimensional space, to analyze the feature space. Figure 6 presents the comparison results of each visualization. Specifically, we randomly selected 200 samples from the four databases and visualized the feature space learned by MADG and the feature space learned by the models in the ablation experiments.

The scatter plot on the right illustrates the feature space learned by MADG, showing that the distribution of feature vectors is more compact and well-separated, indicating that the proposed method is effective in learning discriminative features for face anti-spoofing. In contrast, the plot on the left represents the visualization result of the ablation study without the key component of the multi-domain feature alignment module, $L_{Mdalign}$. In this case, the feature vectors are scattered and poorly separated, highlighting the importance of this module for the proposed method to achieve good performance. Our proposed method achieves better generalization space compared to the model without the multi-domain feature alignment module, and the feature space of real faces is tidier, demonstrating the effectiveness of the proposed method in aligning the feature distribution of real and fake faces.

#### 4.5.2. Analysis of Misclassified Samples

To conduct a comprehensive analysis of our model’s performance, we examined misclassified sample frames, as presented in Figure 7. These samples were taken from the test set of the O&C&I to M task protocol, which is the test data of MSU-MFSD [30]. This dataset includes printed photo and replay video attacks collected from 35 individuals through Android or laptop cameras. The experiments utilized preprocessed face slices.

The left box of Figure 7 displays the misclassified attack samples. In the tests, the misclassified attack faces all belonged to the same individual, and the samples of two attack types obtained by all collection methods from this individual were misclassified. We believe that such errors, which occurred only for attack samples of this individual, are mainly related to individual characteristics.

In contrast, the misclassified real faces in our test involved many individuals, as shown in the right-hand box below. This suggests that our model is less effective in classifying real faces than attacking faces, indicating a certain level of overfitting. This may be for several reasons, including the planar graph architecture of the real face input network with the same dimension as the attack face, the presence of artificial actions in real face videos, such as expressions and motion, and the weakening of sample authenticity due to ornaments and lighting on the faces. Examining classification errors can provide insights for future research.

### 4.6. Conclusion of Experiments

We designed a set of experiments to evaluate the effectiveness of the proposed multi-domain feature alignment framework (MADG). Firstly, we performed an ablation study to assess the contribution of each component to the framework’s classification performance, and our results demonstrate that each component played a significant role in improving performance. Furthermore, we examined the impact of varying weights on the multi-domain alignment process and found that the alignment process of the O&I domain had the greatest interference with the generalization performance in the O&M&I to C protocol. This observation implies that cross-domain alignment between two domains has a significant impact on the overall alignment process due to differences in the feature space. The effectiveness of the alignment module was also demonstrated through comparison. Additionally, we evaluated the performance of MADG on limited source domains and found that the generalization performance of two-source domains was generally inferior to that of three-source domains, indicating the potential of MADG to enhance the classification performance in multi-domain scenarios. Our comparison of different alignment strategies revealed that at least one alignment strategy outperformed the baseline method in most task protocols. Finally, we validated the superiority of MADG by comparing it with the existing SOTA method.

## 5. Conclusions

In this paper we presented a novel approach to improve the generalization capability of face anti-spoofing by utilizing a multi-domain feature alignment domain generalization framework (MADG). To the best of our knowledge, this is the first work to introduce margin discrepancy disparity in domain generalization. Our approach modularizes the alignment algorithm and proposes two multi-domain alignment strategies to enhance the performance of multi-domain alignment. We combine multi-directional triplet loss with the multi-domain alignment module, enabling the effective separation of real and attack face distributions in the feature space. Our method outperforms previous approaches that focus on aligning the entire source domain without considering the distribution characteristics of real and attack faces or only extracting attack face features. We conducted a comprehensive set of experiments on four public databases to validate the effectiveness of our method, and the results demonstrate its superior performance over state-of-the-art approaches.

However, our work also has some limitations that require further improvements in the future. Although our method has made significant progress in the study of feature alignment, it only makes a limited attempt and only involves the exploration of multi-domain problems. Recent studies have shown that the introduction of auxiliary supervision information can effectively address the challenges posed by multi-domain problems. Therefore, future work can be conducted around multi-domain problems and the integration of auxiliary supervision information to further enhance the generalization ability of face anti-spoofing. Overall, our study provides a promising direction for future research on face anti-spoofing.

## Figures and Tables

**Figure 1 sensors-23-04077-f001:**
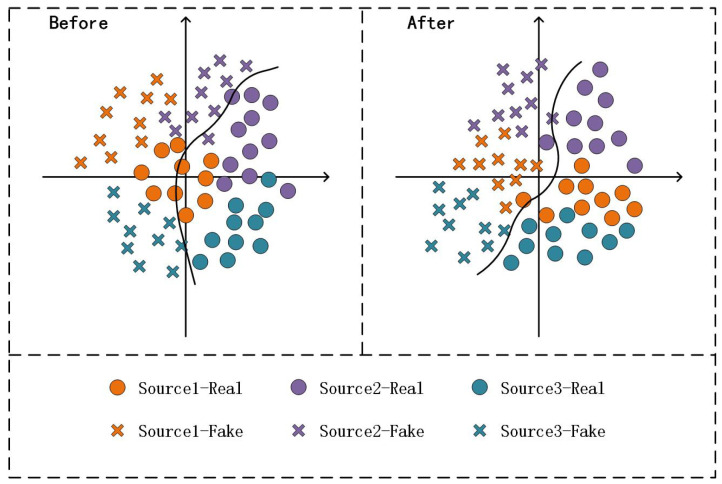
Schematic diagram of the multi-source domain alignment algorithm proposed in this method. The distribution plot on the left illustrates the sample distribution from the source domain before feature alignment, while the plot on the right demonstrates the distribution after feature alignment in the feature space. Notably, the original distributions from different domains exhibit distinct dissimilarities; however, the feature alignment enhances the uniformity of each domain’s distribution in the feature space.

**Figure 2 sensors-23-04077-f002:**
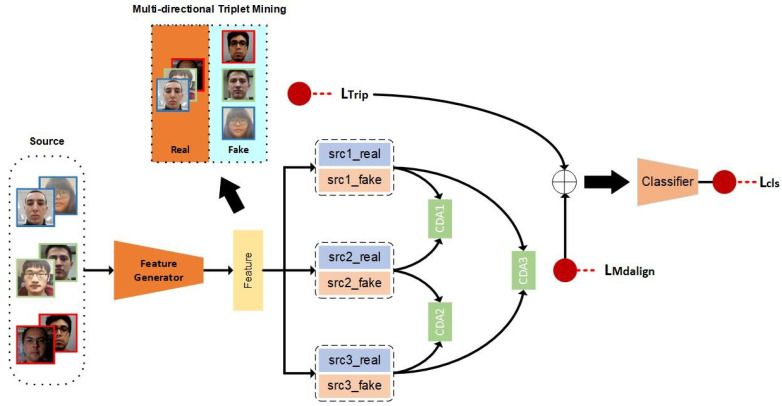
Overview of the proposed method. The proposed method is depicted in the figure, where different domains are delineated by colored borders. The method is exemplified using three source domains. Real and fake face features are obtained from all domains using the same feature generator. These features are then aligned in pairs utilizing a cross-domain alignment (CDA) module, and CDA modules work together to achieve multi-domain feature alignment. Furthermore, multi-directional triplet mining is conducted to disentangle the distribution of real and fake faces in the feature space. The final classifier is trained using cross-entropy loss.

**Figure 3 sensors-23-04077-f003:**
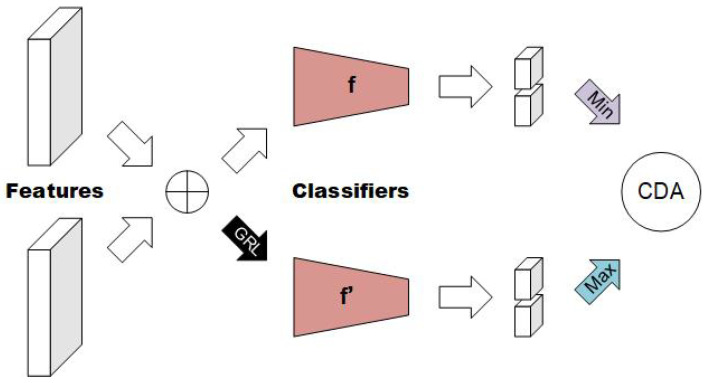
This figure illustrates the cross-domain feature alignment module, which aligns sample features from two domains by minimizing the divergence between them. Adversarial learning is employed to achieve the alignment, where two identical classifiers, denoted as *f* and $f^{\prime }$, are trained to perform the same classification task on the aligned features. Adversarial learning is formulated as a minimax game. During the maximization process, the gradient of the input feature is inverted before being fed to the classifier to ensure the alignment of the two domains. The optimized result is denoted as “CDA” at the end of the process.

**Figure 4 sensors-23-04077-f004:**
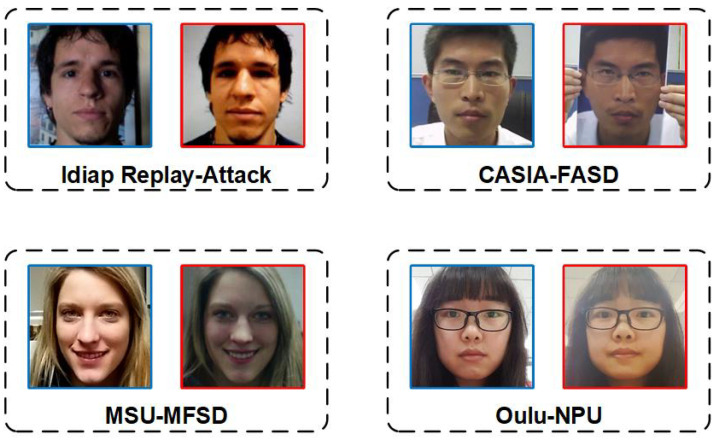
The figure showcases preprocessed sample frames from four publicly available face anti-spoofing databases; namely, Idiap Replay-Attack [28], CASIA-FASD [29], MSU-MFSD [30], and OULU-NPU [31]. Real face samples are demarcated by blue borders, while attack face samples are marked by red borders. The observed variations among the domains are attributed to distinct sampling environmental conditions, emphasizing the need for the design of multiple task protocols to evaluate generalization performance.

**Figure 5 sensors-23-04077-f005:**
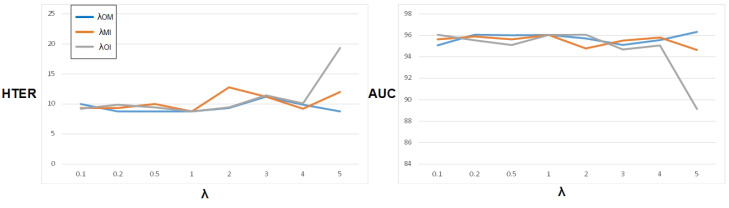
The impact of the alignment weights on the overall generalization performance was evaluated using the O&M&I to C protocol. The x-axis represents the value of λ. λ is the coefficient in the formula $L_{Mdalign}=\frac{2}{N(N-1)}\sum_{i+j=N,i<j}\lambda_{i,j}m_{i,j}$ from Section 3.2 in respect of multi-domain alignment loss, and the control variable method was applied in the experiment, where only one λ was adjusted while the remaining parameters were kept constant. The left figure illustrates variations in the half total error rate (HTER), while the right figure shows the corresponding changes in the area under the curve (AUC).

**Figure 6 sensors-23-04077-f006:**
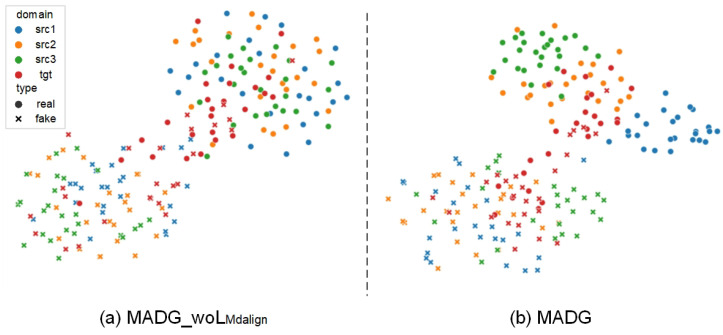
T-SNE visualization was employed to visualize the feature space of high-dimensional data, which enabled the exploration of the relationship between the input features and the output labels. In this study, we selected a subset of 200 samples to visualize the feature distribution. As shown in the scatter plot on the right, our proposed MADG method effectively learns discriminative features that enable clear separation between different domains. In contrast, the visualization of the ablation study, which excludes the multi-domain feature alignment module, exhibits a more disorganized feature distribution. By comparing the visualizations, we can infer that the proposed method achieves better generalization and is effective in aligning the feature distribution of real and fake faces across multiple domains.

**Figure 7 sensors-23-04077-f007:**
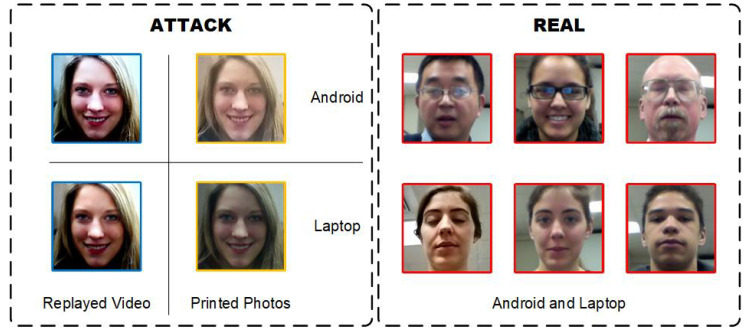
The images depicted above present a collection of misclassified samples obtained from the O&C&I to M task protocol. The left-hand box displays misclassified attack faces, which were all sourced from a single individual whose attack samples were across two types, and all collection methods were misclassified. In contrast, the right-hand box shows some misclassified real face samples, which originated from videos captured via an Android or laptop camera.

**Table 1 sensors-23-04077-t001:** Details of feature generator.

Feature Generator			
Layer	Chan./Stri.	Resampling	Output Shape
Input			
image			
conv1-1	64/2	None	64×128×128
pool1	-/2	-	64×64×64
**layer1**			
conv1-2	64/1	None	
conv2-1	64/1	None	
conv1-3	64/1	None	
conv2-2	64/1	None	64×64×64
**layer2**			
conv1-4	128/2	None	
conv2-3	128/1	None	
conv2-4	128/2	Down	
conv1-5	128/1	None	
conv2-5	128/1	None	128×32×32
**layer3**			
conv1-6	256/2	None	
conv2-6	256/1	None	
conv2-7	256/2	Down	
conv1-7	256/1	None	
conv2-8	256/1	None	256×16×16

**Table 2 sensors-23-04077-t002:** Details of feature embedder and classifier.

Feature Embedder and Classifier			
Layer	Chan./Stri.	Resampling	Output Shape
Input			
Conv2-8			
**layer4**			
conv1-8	512/2	None	
conv2-9	512/1	None	
conv2-10	512/2	Down	
conv1-9	512/1	None	
conv2-11	512/1	None	512×8×8
**bottleneck**			
Average pooling			512×1×1
fc1-1	512/1	-	512
fc1-2	512/1	-	512
fc2-2	2/1	-	2

**Table 3 sensors-23-04077-t003:** Details of alignment classifier.

Alignment Classifier		
Layer	Chan./Stri.	Output Shape
Input		
Conv2-11		
**bottleneck**		
pool	-/1	
fc3-1	1024/1	1024
**head**		
fc4-1	1024/1	
fc5-1	2/1	2
**grl layer**		
		1024
**adv head**		
fc4-2	1024/1	
fc5-2	2/1	2

**Table 4 sensors-23-04077-t004:** Ablation study of different components of the proposed method. The header is the task protocol and the bold represents the optimal result.

Method	O&C&I to M		O&M&I to C		O&C&M to I		I&C&M to O	
	HTER (%)	AUC (%)	HTER (%)	AUC (%)	HTER (%)	AUC (%)	HTER (%)	AUC (%)
$wo_{L_{Mdalign}}$	8.63	95.79	11.22	94.73	16.50	91.42	14.88	92.84
$wo_{L_{Trip}}$	8.40	96.72	**8.56**	**96.95**	22.14	77.60	19.88	87.53
MADG	**7.20**	**97.57**	8.78	96.33	**15.64**	**92.53**	**13.99**	**94.19**

**Table 5 sensors-23-04077-t005:** Experiments on limited source domains involving twelve task protocols. The header is the baseline method and ours; the bold represents the optimal result.

Protocal	SSDG-R [8]		MADG-M (Ours)		MADG-S (Ours)	
	HTER (%)	AUC (%)	HTER (%)	AUC (%)	HTER (%)	AUC (%)
O&M to C	8.11	96.80	8.89	97.04	**7.78**	**97.39**
O&I to C	24.67	84.24	21.89	**88.12**	**20.78**	87.63
M&I to C	**22.67**	**83.54**	23.33	82.32	25.22	81.66
C&M to O	**13.54**	**93.718**	15.09	92.47	15.10	92.35
C&I to O	16.06	92.25	15.42	91.74	**13.87**	**93.17**
M&I to O	**24.17**	**83.28**	**26.29**	**81.50**	27.31	81.19
C&O to M	**6.96**	**97.26**	8.40	96.33	8.87	95.09
C&I to M	11.51	94.91	11.75	94.25	**10.30**	**95.35**
I&O to M	**8.87**	**94.97**	10.31	94.18	9.83	96.72
C&M to I	21.57	85.25	**20.00**	**85.91**	22.21	78.25
C&O to I	23.43	84.22	24.21	83.37	**19.43**	**88.99**
O&M to I	17.14	90.84	**14.43**	**93.96**	19.29	88.80

**Table 6 sensors-23-04077-t006:** Comparisons of different alignment strategies. The header is the baseline method and ours; the bold represents the optimal result.

Protocal	SSDG-R [8]		MADG-M (Ours)		MADG-S (Ours)	
	HTER (%)	AUC (%)	HTER (%)	AUC (%)	HTER (%)	AUC (%)
O&C&I to M	7.38	97.17	7.43	96.40	**7.20**	**97.57**
O&M&I to C	10.44	95.94	**8.78**	**96.33**	9.33	95.35
O&C&M to I	**11.71**	**96.59**	17.14	91.78	15.64	92.53
I&C&M to O	15.61	91.54	**13.99**	**94.19**	15.50	92.76

**Table 7 sensors-23-04077-t007:** Comparison with state-of-the-art methods on limited source domains. The header is the task protocol and the bold represents the optimal result.

Method	M&I to C		M&I to O	
	HTER (%)	AUC (%)	HTER (%)	AUC (%)
MS LBP [9]	51.16	52.09	43.63	58.07
IDA [30]	45.16	58.80	54.52	42.17
LBP-TOP [10]	45.27	54.88	47.26	50.21
MADDG [7]	41.02	64.33	39.35	65.10
SSDG-M [8]	31.89	71.29	36.01	66.88
DR-MD-Net [56]	31.67	75.23	34.02	72.65
ANRL [57]	31.06	72.12	30.73	74.10
SSAN-M [6]	30.00	76.20	29.44	76.62
**MADG (Ours)**	**23.33**	**82.32**	**26.29**	**81.50**

**Table 8 sensors-23-04077-t008:** Comparison with state-of-the-art methods. The header is the task protocol and the bold represents the optimal result.

Method	O&C&I to M		O&M&I to C		O&C&M to I		I&C&M to O	
	HTER (%)	AUC (%)	HTER (%)	AUC (%)	HTER (%)	AUC (%)	HTER (%)	AUC (%)
MS LBP [9]	29.76	78.50	54.28	44.98	50.30	51.64	50.29	49.31
Binary-CNN [15]	29.25	82.87	34.88	71.94	34.47	65.88	29.61	77.54
IDA [30]	66.67	27.86	55.17	39.05	28.35	78.25	54.20	44.59
Color Texture [31]	28.09	78.47	30.58	76.89	40.40	62.78	63.59	32.71
LBP-TOP [10]	36.90	70.80	42.60	61.05	49.45	49.54	53.15	44.09
Auxiliary (Depth)	22.72	85.88	33.52	73.15	29.14	71.69	30.17	77.61
Auxiliary [17]	-	-	28.40	-	27.60	-	-	-
MMD-AAE [49]	27.08	83.19	44.59	58.29	31.58	75.18	40.98	63.08
MADDG [7]	17.69	88.06	24.50	84.51	22.19	84.99	27.89	80.02
PAD-GAN [56]	17.02	90.10	19.68	87.43	20.87	86.72	25.02	81.47
RFM [52]	13.89	93.98	20.27	88.16	17.30	90.48	16.45	91.16
SSDG-M [8]	16.67	90.47	23.11	85.45	18.21	94.61	25.17	81.83
SSDG-R [8]	7.38	97.17	10.44	95.94	**11.71**	**96.59**	15.61	91.54
ANRL [57]	16.03	91.04	10.83	**96.75**	17.85	89.26	15.67	91.90
DRDG [58]	15.56	91.79	12.43	95.81	19.05	88.79	15.63	91.75
**MADG (Ours)**	**7.20**	**97.57**	**8.78**	96.33	15.64	92.53	**13.99**	**94.19**

## Data Availability

The data presented in this study are openly available in [repository name CASIA-FASD] at [http://www.cbsr.ia.ac.cn/english/FASDB_Agreement/Agreement.pdf], ref. [29]; [repository name Idiap Replay-Attack] at [https://www.idiap.ch/en/dataset/replayattack], ref. [28]; [repository name MSU-MFSD] at [http://biometrics.cse.msu.edu/Publications/Databases/MSU%20Mobile%20Face%20Spoofing.rar], ref. [30]; [repository name OULU-NPU] at [https://www.sites.google.com/site/oulunpudatabase/], ref. [31] (all accessed on 1 February 2023).

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
