# Peer review of "Multi-Domain Feature Alignment for Face Anti-Spoofing"

_sensors, 2023, doi:10.3390/s23084077_

Round 1

Reviewer 1 Report

Authors try to propose a method of source domain alignment based on margin disparity discrepancy and adversarial learning. Some interesting results are provided in the manuscript. Howerever, there are some problems in this manuscript.

1. The manuscript with a combination of existing methods lacks  sufficient  theoretical innovation.

2. The main contributions of this work should be refined.

3. Some figures should be improved. For exampe  Figure 2 should be  smaller font size and Compact layout; Figure 5 is not clear.

4. Is the data of other methods in Table 7 from the their references or the experiment done by yourself.

5. The methods used for comparison are generally older, and many new methods should be added.

Author Response

Thank you for your insightful review of our manuscript. We appreciate the time and effort you have invested in providing us with valuable feedback and suggestions. We have carefully considered your comments and incorporated revisions to the manuscript accordingly.

Regarding the theoretical novelty of our approach, we want to emphasize that to the best of our knowledge, we are the first to apply margin disparity discrepancy to domain generalization. Additionally, we have introduced a novel feature alignment technique that employs adversarial neural networks and two alignment strategies to enhance multi-source domain feature alignment efficacy. We have also modularized the alignment algorithm and introduced a multi-directional triplet mining algorithm that sharpens the feature space's boundaries and boosts classification performance.

We appreciate your suggestion to refine the main contributions of our work and have updated the manuscript to more clearly state the contributions of our proposed method, which can now be found in line 85. Our contributions include the introduction of the cross-domain feature alignment algorithm, multi-domain feature alignment strategies, the Multi-domain Feature Alignment Domain Generalization (MADG) framework, and the experimental validation of our proposed method.

Regarding the figures in the manuscript, we have made adjustments as you suggested. Specifically, we have improved the font size and layout of Figure 2 to make it more compact and enhanced the clarity of Figure 5.

We want to clarify that the data presented in Table 7 are from references of the methods we compared in our experiments, organized according to top conference standards (i.e., CVPR, ECCV) and top journals (i.e., tpami, tifs). We have appropriately cited the comparison methods and reproduced SSDG, SSAN, and other methods with results that are consistent with those reported in their original papers.

We acknowledge your comment regarding the age of some of the compared methods. To address this, we have included the latest state-of-the-art methods, such as SSAN from CVPR2022, in our comparison, in addition to selecting other methods published in top conferences (i.e., CVPR, ACMMM) and journals (i.e., tpami, tifs) (can be found in Table 8).

Finally, we have made comprehensive changes to the language of the manuscript to improve its clarity and readability. Thank you again for your valuable feedback, which has helped us to enhance the quality of our work.

Once again, we would like to express our gratitude for your valuable comments and suggestions. Your feedback is invaluable to us in improving the quality of our paper.

Reviewer 2 Report

Introduction:
It is necessary in the introduction to locate the reader in each of the sections of the paper.

Identify beyond the implementation of the method what is the problem to be solved.
Conclusions:
It is necessary to complement the conclusions, showing in more depth the contribution of the method shown.
To strengthen the bibliographical references I recommend the inclusion of the following references:

Liu, Z., Gu, X., Chen, J., Wang, D., Chen, Y., & Wang, L. (2023). Automatic recognition of pavement cracks from combined GPR B-scan and C-scan images using multiscale feature fusion deep neural networks. Automation in Construction, 146, 104698.

Ariza-Colpas, P., Piñeres-Melo, M., Barceló-Martínez, E., De la Hoz-Franco, E., Benitez-Agudelo, J., Gelves-Ospina, M., ... & Leon- Jacobus, A. (2019). Enkephalon-technological platform to support the diagnosis of Alzheimer's disease through the analysis of resonance images using data mining techniques. In Advances in Swarm Intelligence: 10th International Conference, ICSI 2019, Chiang Mai, Thailand, July 26–30, 2019, Proceedings, Part II 10 (pp. 211-220). Springer International Publishing.

Author Response

We would like to express our gratitude for your valuable feedback on our paper. Your constructive comments have been invaluable in helping us to improve the quality of our manuscript. We are pleased to inform you that we have carefully considered your suggestions and made several revisions to the introduction and conclusion sections.

In response to your feedback, we have added a paragraph to the introduction that outlines the content of each section (see line 101) and provides a clear statement of the problem our method addresses (see line 65). We have also provided a comprehensive summary of our experiments in subsection 4.6 (see line 627) and have emphasized the theoretical innovation of our method in the conclusion, along with a more detailed discussion of potential avenues for future research (see line 645). Furthermore, we have refined the full-text language to improve clarity and readability.

We would also like to thank you for providing additional references for us to consider. We have reviewed these papers and have incorporated them into our references.

Once again, thank you for your valuable comments and suggestions. If you have any further comments or suggestions, please do not hesitate to let us know. Your feedback is greatly appreciated and has helped us to improve the quality of our paper.

Reviewer 3 Report

This paper introduces an adversarial technique to perform feature alignment across multiple domains and improve the performance of face anti-spoofing models. The introduction discusses the motivation and research questions clearly and the related work section shows that the authors know the state of the arts in the domain. The proposed method section is well-written such that it has been started with an overview of the model and then each component has been explained. The evaluation of the proposed method has been done on 4 datasets and the results have been compared with several existing works. The evaluation has been extended by conducting some ablations studies. Overall, the paper is carefully written and could be published after a minor revision:

- Section 2.1 misses a basic discussion about the face anti-spoofing methods and how they work. 

- Tables should be self-explanatory, by extending the captions.

Author Response

Thank you for providing valuable feedback on our paper. We appreciate your time and effort in providing us with constructive criticism. We are glad to know that you found our work well-written and carefully executed.

We have carefully considered your feedback and made the following revisions:

We added a brief discussion in Section 2.1 about face anti-spoofing methods and how they work to provide better context for readers (see line 137).
We extended the captions of the tables to make them more self-explanatory.
Furthermore, we have refined the full-text language to improve clarity and readability.

Once again, thank you for your suggestions. Your feedback will help us improve the quality of our paper. If you have any further comments or suggestions, please do not hesitate to let us know.

Round 2

Reviewer 1 Report

The  manuscript has been improved well.